# Functional convalescent plasma antibodies and pre-infusion titers shape the early severe COVID-19 immune response

Jonathan D. Herman [1,2,8], Chuangqi Wang[3,8], Carolin Loos[1,3,8], Hyunah Yoon [4], Johanna Rivera[4,5], M. Eugenia Dieterle [5], Denise Haslwanter[5], Rohit K. Jangra[6], Robert H. Bortz III[5], Katharine J. Bar[7], Boris Julg [1], Kartik Chandran [5], Douglas Lauffenburger [3✉], Liise-anne Pirofski[4,5✉] & Galit Alter [1✉]

Transfer of convalescent plasma (CP) had been proposed early during the SARS-CoV-2 pandemic as an accessible therapy, yet trial results worldwide have been mixed, potentially due to the heterogeneous nature of CP. Here we perform deep profiling of SARS-CoV-2-specific antibody titer, Fc-receptor binding, and Fc-mediated functional assays in CP units, as well as in plasma from hospitalized COVID-19 patients before and after CP administration. The profiling results show that, although all recipients exhibit expanded SARS-CoV-2-specific humoral immune responses, CP units contain more functional antibodies than recipient plasma. Meanwhile, CP functional profiles influence the evolution of recipient humoral immunity in conjuncture with the recipient's pre-existing SARS-CoV2-specific antibody titers: CP-derived SARS-CoV-2 nucleocapsid-specific antibody functions are associated with muted humoral immune evolution in patients with high titer anti-spike IgG. Our data thus provide insights into the unexpected impact of CP-derived functional anti-spike and anti-nucleocapsid antibodies on the evolution of SARS-CoV-2-specific response following severe infection.

[1] Ragon Institute of MGH, MIT, and Harvard, Cambridge, MA, USA. [2] Division of Infectious Disease, Brigham and Women's Hospital, Boston, MA, USA. [3] Department of Biological Engineering, Massachusetts Institute of Technology, Cambridge, MA, USA. [4] Division of Infectious Diseases, Department of Medicine, Albert Einstein College of Medicine and Montefiore Medical Center, Bronx, NY, USA. [5] Department of Microbiology and Immunology, Albert Einstein College of Medicine, Bronx, NY, USA. [6] Department of Microbiology and Immunology, Louisiana State University Health Science Center-Shreveport, Shreveport, LA, USA. [7] Department of Medicine, University of Pennsylvania, Philadelphia, PA, USA. [8] These authors contributed equally: Jonathan D. Herman, Chuangqi Wang, Carolin Loos. ✉email: lauffen@mit.edu; l.pirofski@einsteinmed.org; galter@mgh.harvard.edu

The emergence of the novel coronavirus, SARS-CoV-2 and the resultant disease COVID-19 caused an international pandemic unseen since the 1918 influenza pandemic[1] that led to worldwide lockdowns, declining economies, overburdened health systems, and nearly 4 million deaths as of July 2021[2,3]. Severe COVID-19, which develops in 14% of the infected population[4], can lead to acute respiratory distress syndrome, renal failure, thromboembolic complications, a hyperinflammatory syndrome, and death[5–7]. While successful vaccine development has progressed at an unprecedented speed[8], there is a paucity of proven therapies for hospitalized patients with severe COVID-19. The need for effective therapies for such patients is highlighted by the slow pace of the global vaccine rollout and the continued emergence of viral variants of concern[9–11] highlight the need for effective therapies for COVID-19.

COVID-19 convalescent plasma (CP) was proposed as a possible therapeutic[12] early in the pandemic because of its antiviral activity, plausible biological mechanism of action, and its use in epidemics when no other therapies were available[12] including the 1918 flu[13], SARS[14,15], and H1N1[16,17]. Conceptually, CP would exert an antiviral effect by providing anti-SARS-CoV-2-specific antibodies that could neutralize the virus, blunt viral replication, and thereby prevent viral dissemination and subsequent damage. Now, more than a year into the COVID-19 pandemic, CP has been shown to be a safe intervention[18], but the optimal patient and clinical stage of COVID-19 illness for the administration of this potential therapeutic agent is unclear.

Randomized control trials (RCT) suggest that early administration of CP to patients with COVID-19 may confer a benefit in hospitalized patients[19]. Although CP has not improved overall clinical outcomes when given to patients with severe (requiring oxygen supplementation) or life-threatening (requiring ICU care or mechanical ventilation) COVID-19[20–22], a mortality benefit was observed in one small RCT in which patients were treated later in disease[23]. In addition, CP has consistently demonstrated an antiviral effect and was associated with improved inflammatory markers even in studies with no overall benefit[20–22]. Although the "active" agent in CP is considered to be SARS-CoV-2 Spike antibodies that neutralize the virus[19–21,24,25], there is a gap in our knowledge of the therapeutic role of SARS-CoV-2 Fc-effector functions in CP.

Each unit of CP is obtained from a single individual[26] and may differ more broadly based on the CP donor's genetics[27–29], severity of antecedent COVID-19 illness[30–32], and time since recovery from COVID-19[32,33]. While the mixed results of CP study outcomes may in part be due to a lack of standardization of antibody titer and neutralization assays used to select CP, we hypothesized that CP Fc-effector mediated functional activity may contribute to differences in the impact of CP on recipient immune profiles and a Systems Serology[34] approach would elucidate this.

Here, we use systems Serology to comprehensively and agnostically analyze SARS-CoV-2 antibody profiles in CP units and the corresponding CP recipients. We measure Spike- and Nucleocapsid-specific antibody titers, Fc-receptor binding, and Fc-driven antibody functions, including antibody-dependent complement deposition (ADCD), antibody-dependent cell phagocytosis (ADCP), antibody-dependent neutrophil phagocytosis (ADNP), and antibody-dependent NK cell activation (ADNK), in plasma from 19 CP-treated patients hospitalized with severe COVID-19 and the CP units they received. Despite the heterogeneity in CP antibody profiles, CP units harbor more functional antibody activity than CP recipient plasma. Further, CP administration blunts the evolution of inflammatory humoral immune profiles via distinct mechanisms depending on pre-CP SARS-CoV-2-specific antibody titers in the recipients. These results suggest that the CP and other antibody-based therapies may provide benefits beyond simple pathogen neutralization and attenuation, via the inflammatory humoral immune responses that are associated with severe COVID-19 disease.

## Results

**Antibody profiles of COVID-19 CP.** The role of Fc-effector functions in both resolution of COVID-19[30] and vaccine-induced immunity[35,36], suggests that non-neutralizing antibody activity may have critical therapeutic effects, that could also influence CP efficacy. We performed Systems Serology analyses on pre- and post-CP administration plasma samples from a previously reported cohort of hospitalized CP recipients and the CP units they received[37]. A cohort of 19 severely ill COVID-19 patients, who received CP within 72 h of hospital admission from April 13th to May 4th 2020 in the Bronx, NY, and the CP units they received were profiled in this study (Supplementary Table 1)[37]. All patients required non-invasive oxygen supplementation. At the time of study enrollment, they had a median score of 5 on the 8-point World Health Organization Ordinal Scale for clinical improvement[38], indicating they were hospitalized and required non-invasive ventilation or high-flow nasal cannula. However, 48% subsequently required non-invasive positive pressure ventilation or mechanical ventilation during the course of this study. All patients received 1 unit of 200 mL CP pre-screened for Spike IgG titer by ELISA within 3 days of hospitalization and were evaluated for clinical status and laboratory measures up to 28 days after enrollment (as described in Yoon et al.[37]). By day 28 of the study, five patients died, one patient remained hospitalized, and 13 were discharged (Supplementary Table 1).

CP units used in the study were obtained from donors who recovered from mild COVID-19 and were never hospitalized[37]. Antibody profiling was performed on 18 of 19 CP units (obtained from 14 different donors) and patient plasma was obtained pre-CP (day −1), 1 day after CP (day 1), and 3 days after CP (day 3) as depicted in Supplementary Fig. 1A. CP units from four donors were each given to two unique patients. CP units from the 10 other donors were each given to a single patient. The CP unit for one patient was not available. The majority of patients, 17 of 19, were treated with corticosteroids during their hospitalization. Two patients received CP prior to corticosteroids. Patients enrolled in this study received corticosteroids for a median of 5 days (Supplementary Table 1). Only one patient each received the following other COVID-19 investigational therapies at the time of the trial: remdesivir, sarilumab, and leronlimab.

Initial analysis of SARS-CoV-2-specific antibody profiles in CP units is shown in Fig. 1a. While Spike (S)-specific IgG1 was detected in all CP units, IgG-subclass distribution varied. CP units 10 and 14 had an IgG3-centric response, while CP units 6, 7, and 9 had an expanded IgA response. Neutralizing antibody titers (NT50) varied from 59 to 12,400 among the CP donors, with CP units 12, 13, and 14 having the highest titers. The Fc-receptor (FcR) binding also varied substantially, with FCR2A binding expanded in CP units 7, 10, and with expanded binding to all FcRs in CP unit 14. Fc-effector functions were present in all CP units, with the broadest levels of functions noted in CP units 1, 5, 7, and 10. Only low-level correlations were found between neutralization titer and the S-specific Fc profiles (Fig. 1b and Supplementary Fig. 1B, C). Collectively, these data point to the heterogeneity of CP not only in antibody titer but also in Fc-directed antibody functional capacity.

**Enrichment of anti-SARS-CoV-2 functional antibodies in CP.** Specific differences in CP unit and CP recipient antibody profiles are poorly understood. We compared the pre-CP administration

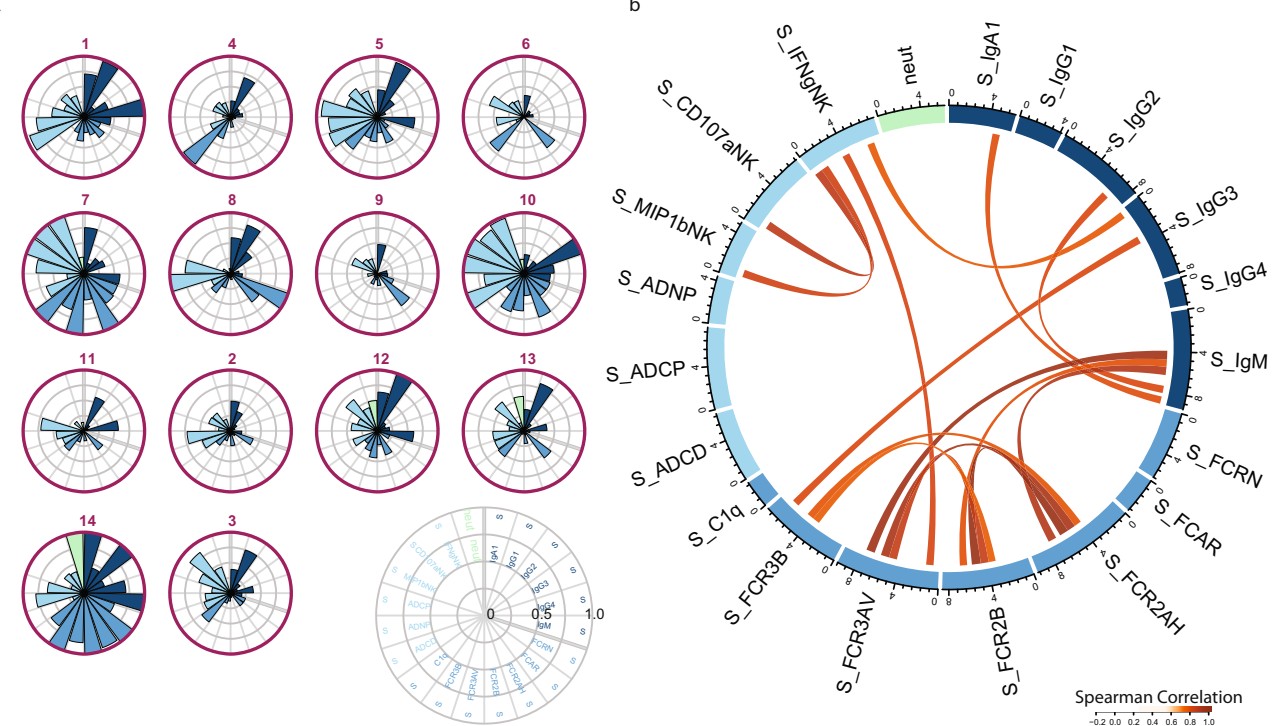

**Fig. 1 Heterogeneity of anti-spike antibody profiles in convalescent plasma was not captured by neutralizing antibody titer.** CP units were profiled for SARS-CoV-2S-specific antibody responses. **a** Each polar plot depicts an individual donor's anti-Spike antibody profile, scaled to the minimum and maximum of the 14 CP units profiled. Each wedge represents a SARS-CoV-2 antibody feature, and the size of the wedge indicates the magnitude of the value. The colors represent the feature group: dark blue—antibody isotypes and subclasses; blue—Fc-receptor binding levels; light blue—antibody-dependent functions; light green—neutralizing antibody titer. **b** Cord diagram representing the statistically significant correlation among S-specific antibody features in CP units with Spearman correlation > 0.5 and two-sided *p*-value < 0.05. The width of antibody each feature represents the accumulated values of Spearman correlation coefficients of that feature with all other features included in the diagram. The color of the cord represents the strength of Spearman correlation, with a higher correlation having darker red values. Source data are provided as a Source Data file.

(day −1) plasma profiles of CP recipients to the CP units they received (Fig. 2a). While variation was observed across CP recipient profiles (Supplementary Fig. 2A, B), pre-CP recipient plasma exhibited higher anti-SARS-CoV-2 titers and FcR binding than CP (Fig. 2A). IgA titers and Fcα-receptor (FCAR) binding were most significantly expanded in CP recipients compared to CP units (Fig. 2a, d and Supplementary Fig. 2C). Conversely, Fc-mediated effector functions were markedly more robust in CP, despite the lower S-specific subclass and isotype titers, and FcR-binding. These observations suggested that units of CP that are collected following the resolution of mild SARS-CoV-2 infection, may have qualitatively different antibodies with superior function.

To fully capture the differences between CP and recipient antibody profiles, we performed a partial least squares discriminant analysis (PLS-DA) (Fig. 2b and Supplementary Fig. 2D, E). The PLS-DA model separated the two clinical groups based on SARS-CoV-2 antibody profiles, that were statistically significantly different based on a permutation test (Fig. 2B and Supplementary Fig. 2F). Among the top features that distinguished the groups, six features were enriched in CP and four features were enriched in the CP recipients. CP selectively exhibited enhanced levels of multiple measures of antibody function including ADCP, ADNK (S_ MIP-1βNK, S_CD107aNK), and ADCD (Fig. 2C), which were variably enriched in CP recipient plasma (Supplementary Fig. 2E). Conversely, IgM titers and IgA-binding to the IgA-Fc-receptor (FCAR) responses were selectively expanded in CP recipient plasma compared to CP units (Fig. 2C).

Given the highly correlated nature of the polyclonal SARS-CoV-2 antibody response, we sought to identify whether additional antibody features tracked with the top antibody features. Two

distinct polyclonal networks were observed in Fig. 2d. The larger network included all CP-enriched functional features and CP recipient-enriched RBD-specific IgM, proximally linked to S-specific ADCD, pointing to an enrichment of complement activity in CP and recipient antibodies. The smaller network of largely IgA and FCAR features was enriched in CP recipient plasma features (Fig. 2d), pointing to an exclusive IgA centric-signature in COVID-19 patients.

**Impact of CP on recipient SARS-CoV-2-specific antibody profiles.** Given the differences in CP unit and CP recipient plasma, we next investigated the effect of CP administration on the overall evolution of the humoral immune response in CP recipients. As shown in Fig. 3a, most recipients exhibited an increase in Spike-specific IgG1 (S-IgG1) titers 1 and 3 days after CP administration. There was also a global increase of S-specific subclass, isotype, FcR binding, and functional responses was observed as early as day 1 and later at day 3 post-CP administration (Fig. 3b and Supplementary Fig. 3A). To identify the features that changed most over time, we used multivariate models to compare CP recipient plasma antibody profiles pre-CP to day 1 (Fig. 3c) and day 3 (Supplementary Fig. 3D) after CP treatment. Significant differences were noted in the multivariate models marked by a largely expanded humoral immune response, with the evolution of less functional antibody subclasses including IgG4 and IgG2 to several antigens including to the receptor-binding domain (RBD), S, and S1 (Supplementary Fig. 3B–E). Conversely, S1-specific FCAR binding and N-specific antibody binding to the neonatal FcR (FcRn) were lost over time

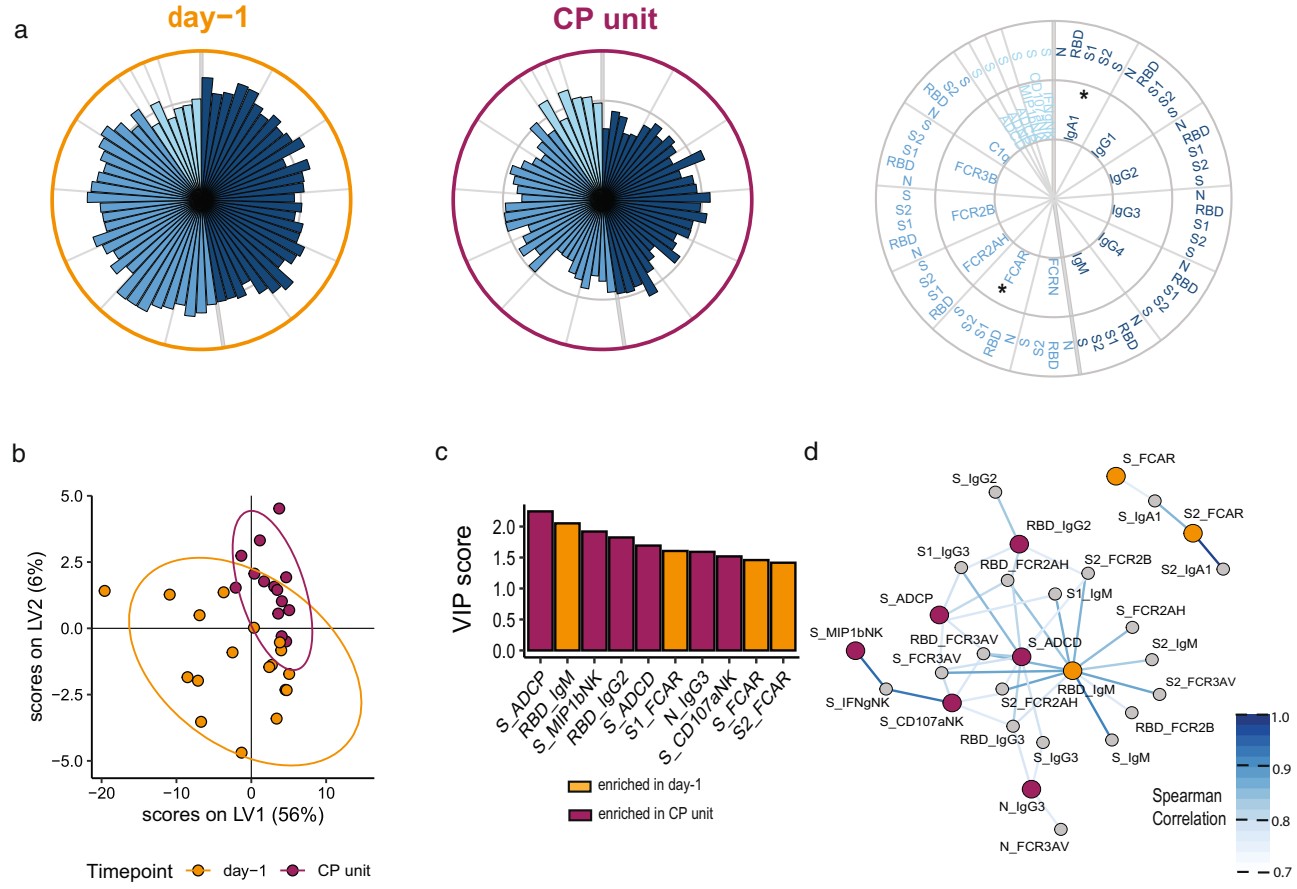

**Fig. 2 Functional antibodies were enriched in convalescent plasma.** CP recipients and matched CP units were profiled for SARS-CoV-2-specific antibody responses. **a** The polar plots depict the mean percentiles of SARS-CoV-2-specific antibody features within the day −1 (*n* = 18) and CP unit (*n* = 14) groups for the antigens N, RBD, S1, S2, and S. Each wedge represents a SARS-CoV-2 antibody feature, and the size of the wedge indicates the magnitude of the value. The colors represent the feature group: dark blue—antibody isotypes and subclasses; blue—Fc-receptor binding levels; light blue—antibody-dependent functions. An asterisk indicates a global *p*-value obtained by non-parametric combination (two-sided Mann–Whitney *U* test *p*-values for partial tests within each feature type, and Fisher's method for combination, *$p < 0.05$). Partial least squares discriminant analysis (PLS-DA) was applied to identify features distinguishing convalescent from day −1 severe COVID-19 patients. **b** PLS-DA scores plot for the first two latent variables. Each dot is one sample, and the ellipses indicate 95% confidence regions assuming a multivariate *t*-distribution. The model achieved an average cross-validation accuracy of 72%. **c** Variable importance in projection (VIP) score plot showing the top 10 important antibody features out of the 81 antibody features used to construct the PLS-DA model. The color of the bar indicates in which group the feature is enriched, i.e., has a higher median value. **d** Co-correlation network of features from the PLS-DA model in **b** with the top 10 VIP scores. The color of the nodes indicates whether the feature was enriched in donor CP vs. day −1 severe COVID-19 patients. Nodes that were not top features were colored gray. Nodes enriched in day −1 were colored in orange and nodes enriched in donor CP were colored in maroon. Edges were included if Spearman correlation was statistically significant after Benjamin–Hochberg correction of a two-sided *$p < 0.05$* and the value of *R* coefficient was above 0.75. The strength of the correlation was represented by the color of the edge with higher correlation coefficients represented by darker blue. Source data are provided as a Source Data file.

(Supplementary Fig. 3B,C). Finally, we examined the relationship between CP antibody functions and the evolving humoral immune response in the CP recipients (Fig. 3e and Supplementary Fig. 3F). Overall, CP-derived antibodies with strong S-ADNK, measured by chemokine (MIP-1β) and degranulation (CD107a), were correlated with increasing S-specific recipient antibody responses. On the other hand, CP-derived antibodies with high complement (ADCD) and phagocytic (ADCP) activity were associated with decreasing S-specific recipient antibody responses (Fig. 3e). These data suggest that CP-mediated ADCP and ADCD may dampen the evolution of the S-specific response in CP recipients.

**Pre-existing spike IgG1 titers determined the effect of CP on severe COVID-19 patients.** Clinical data suggest that pre-existing

antibody levels may affect the therapeutic benefit of transferred anti-Spike monoclonal antibodies and CP[39]. To ascertain whether pre-existing antibody titers in recipients shaped CP effects, we separated the CP recipients into those with pre-transfusion high or low S-IgG titers (Fig. 4a). SARS-CoV-2 antibody evolution was observed in both the low and high pre-existing S-IgG cohorts (Fig. 4a). Importantly, although these groups differed in their pre-transfusion S-IgG titers, they had symptomatic COVID-19 for similar amounts of time (Supplementary Fig. 4a) prior to CP treatment. This suggested that the pre-transfusion S-IgG titers were a marker of a quality of the patient's humoral immune response, rather than just a surrogate of COVID duration.

To define the role of CP in shaping the evolution of the humoral immune response, we compared CP antibody profiles to the change in antibody features from day −1 to day 1 in the

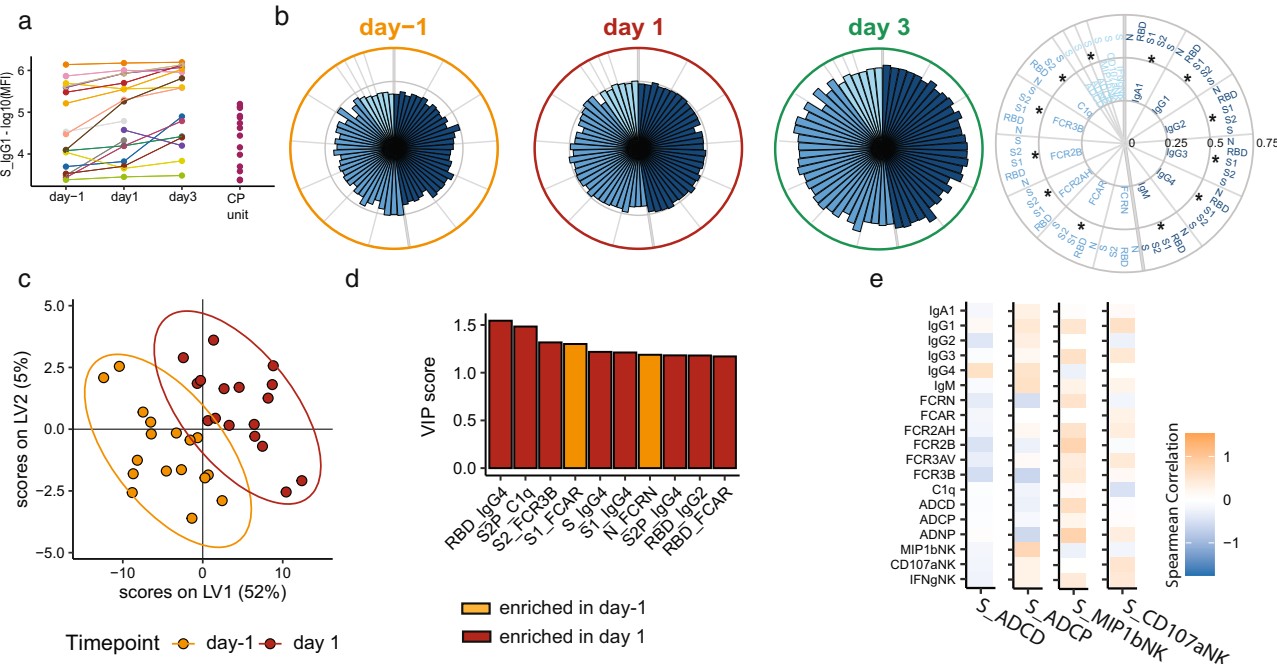

**Fig. 3 SARS-CoV-2-specific antibody profiles globally increased in severe COVID-19 CP recipients.** CP recipients were profiled for SARS-CoV-2-specific antibody responses from 1 day before receiving CP (day −1) to 1 and 3 days after receiving CP. **a** SARS-CoV-2S-specific IgG1 titers of COVID-19 patients at day −1 ($n = 18$), day 1 ($n = 18$), and day 3 ($n = 16$), and 14 CP units. Values are reported as log10 median fluorescence intensity (MFI). **b** The polar plots depict the mean percentiles of SARS-CoV-2-specific antibody features within the day −1 ($n = 18$), day 1 ($n = 18$), and day 3 ($n = 16$) groups for the antigens N, RBD, S1, S2, and S. Each wedge represents a SARS-CoV-2 antibody feature, and the size of the wedge indicates the magnitude of the value. The colors represent the feature group: dark blue—antibody isotypes and subclasses; blue—Fc-receptor binding levels; light blue—antibody functions. An asterisk indicates a global two-sided $p$-value obtained by non-parametric combination (Friedman test $p$-values for partial tests within each feature type, and Fisher's method for combination, *$p < 0.05$). **c** Multi-level partial least squares discriminant analysis (mPLS-DA) scores plot for the first two latent variables. Each dot is one sample, and the ellipses indicate 95% confidence regions assuming a multivariate $t$-distribution. Colors indicate the time point the samples of the $n = 17$ individual patients were taken (day −1 and day 1). The model achieved an average cross-validation accuracy of 86%. **d** Variable importance in projection (VIP) score plot showing the top 10 important antibody features out of the 81 antibody features used to construct the mPLS-DA. The color of the bar indicates in which group the feature is enriched, i.e., has a higher median value. **e** Heatmap showing the Spearman correlation coefficients between increases in antibody levels between day −1 and day 1 ($y$ axis) and corresponding donor CP Spike-specific antibody features enriched in CP including ADCD, ADCP, and ADNK activation measured by NK cell expression of MIP-1β and CD107a. Source data are provided as a Source Data file.

plasma of CP recipients with high and low pre-existing S-IgG, respectively (Fig. 4b, c). CP profiles had distinct effects on CP recipient antibody evolution depending on their pre-existing S-specific titers. While most features were largely positively correlated across the CP units and the CP recipients, N-specific CP antibody features were largely negatively correlated with all SARS-CoV-2-specific evolutionary profiles in the high S-specific IgG titer group (Fig. 4b). Conversely, more variable relationships were observed in the low S-specific IgG titer group (Fig. 4b).

Next, we identified the CP-specific features that differentially influenced recipient profile evolution by calculating the difference between the high (Fig. 4b) and low (Fig. 4c) day −1 titer correlation matrices, displayed in Supplementary Fig. 4B. Next, the median value for each CP antibody feature (each column of Supplementary Fig. 4B) was calculated and permutation testing was then performed to determine whether the median differences were statistically significant (Fig. 4d). This analytic approach confirmed the unique relationship between the S- and N-specific functional antibodies modulate the evolution of humoral responses in the recipients from day −1→1 depending on pre-existing S-IgG1 titer (Fig. 4d).

Next, we performed an analysis of the CP features that were most strongly associated with day −1→1 changes that differed between recipients with high or low IgG titers prior to CP administration. The influence of each CP feature on the trajectory of each recipient antibody feature (positive vs. negative correlation with day 1–day −1) was interrogated in the high and low IgG titer recipients (Fig. 4e, f). S1-IgG3 and N-IgG3 titers in CP units were associated with attenuation of the inflammatory antibody profiles in CP recipients with high titers on day 1 (Fig. 4e) and day 3 (Supplementary Fig. 5F). Conversely, S-ADNP was highly associated with limiting inflammatory antibody features only in CP recipients with low day −1 S-IgG titers (Fig. 4f and Supplementary Fig. 5F). Collectively, these data highlighted the unexpected and differential effect of donor CP on modulating endogenous SARS-CoV-2 antibody evolution, resulting in both pro- and anti-inflammatory effects depending on pre-existing antibody levels in patients with severe COVID-19.

## Discussion

In this study, we used system serology to characterize the SARS-CoV-2 isotype, subclass, FcR binding, and Fc-mediated functional activity of CP units administered to 19 hospitalized COVID-19 patients and pre- and post-CP administration plasma of the recipients (Supplementary Fig. 1A). Our data show that while the overall profiles of CP were heterogeneous, CP was highly enriched with Fc-functional antibodies. In contrast, CP recipient profiles were characterized by expanded IgG and IgA responses, with more limited Fc-effector profiles. Moreover, CP Fc-

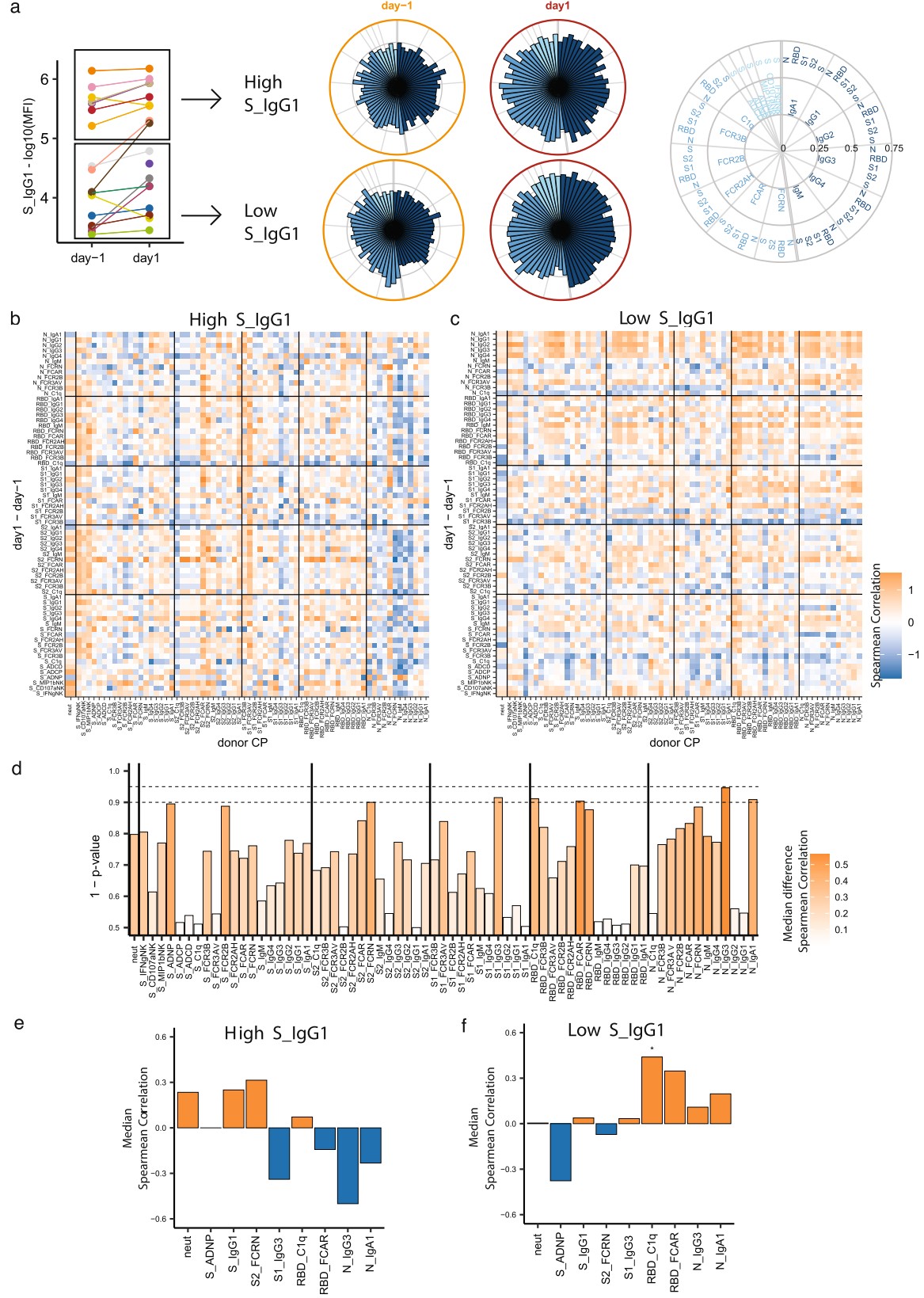

functional antibodies appeared to influence the evolution of CP recipient antibody responses, by limiting the emergence of inflammatory antibody signatures in both recipients with high and low pre-CP-S-specific IgG titers, albeit through different mechanisms. Overall, our findings provide new insights into immunomodulatory functions of CP that may help explain its apparent benefit in some patients with severe disease, and its ability to clear virus and reduce inflammatory markers even when it does not affect overall clinical outcome.

Pathogen neutralization has been traditionally regarded as the key mechanism by which antibodies may confer protection and therapeutic benefit[35,40]. Though many lines of evidence suggest

**Fig. 4 Effect of convalescent plasma on recipient antibody trajectory was influenced by recipient pre-existing spike IgG1 titers.** The effect of CP antibody features on the trajectory (day 1 vs day −1) of the SARS-CoV-2 humoral response was profiled separately for CP recipients with high and low pre-existing S-IgG1 antibodies. **a** SARS-CoV-2S-specific IgG1 titers of CP recipients at day −1 (n = 18) and day 1 (n = 18). Patients were separated by the mean day −1 S-IgG1 titer. Values are reported as log10 MFI. The polar plots depict the mean percentiles of SARS-CoV-2-specific antibody features within the day −1 (n = 18), day 1 (n = 18) groups for the antigens N, RBD, S1, S2, and S in the High and Low S-IgG1 groups. Each wedge represents a SARS-CoV-2 antibody feature, and the size of the wedge indicates the magnitude of the value. The colors represent the feature group: dark blue—antibody isotypes and subclasses; blue—Fc-receptor binding levels; light blue—antibody functions. **b**, **c** Heatmap showing the Spearman correlation coefficients between the trajectory of patient's antibody levels between day −1 and day 1 (y axis) and corresponding CP unit antibody features (x axis) in patients with high (**b**) and low (**c**) levels of pre-existing S-IgG1 antibody. **d** Bar plot representing the statistical significance of CP-derived antibody features that have differential effects on SARS-CoV-2 humoral response depending on the pre-existing Ab titers. The Color represents the absolute value of the median difference of correlation in the high and low pre-existing Ab groups, with dark orange representing higher absolute differences. The height of the bar represents the two-sided p-value determined by the permutation test. The two dotted lines represent a p-value of 0.01 and 0.05 respectively. **e**, **f** Bar plots of the feature in **d** with p-value ≤ 0.1, S-IgG1 and Neutralizing antibody titer. The color of the bars represents the median Spearman correlation of CP features in high pre-existing S-IgG1 (**e**) and low pre-existing S- IgG1 (**f**). The orange and blue coloring of the bars represents positive and negative Spearman correlation respectively. An asterisk represents a CP donor antibody feature with a significant global two-sided p-value obtained by non-parametric combination (permutation test p-values for partial tests within each paired feature, and Fisher's method for combination, *p < 0.05) within high (**e**) and low (**f**) pre-existing S-IgG1 patients. Source data are provided as a Source Data file.

neutralizing antibodies are important for prophylactic protection from COVID-19[41,42], neutralization has not been linked to the natural resolution of infection[30], and instead has been associated with disease progression[43]. However, the therapeutic benefit in COVID-19[44] of IVIG, a pool of immunoglobulins collected from thousands of healthy donors collected before the COVID-19 pandemic, raises the possibility that antibodies may provide protection via an alternative, non-neutralizing, and even non-pathogen-specific mechanisms[45]. Emerging data suggest CP can modulate immunity, resulting in decreased cytokines in severe COVID-19[46]. In addition, antibodies can drive rapid clearance of virally infected cells via complement activation, cytotoxic destruction, or opsonization of the virus or infected cells[47]. Along these lines, here we observed a role for both S- and N-specific antibody Fc-properties in attenuating the evolution of inflammatory humoral immune responses. Thus, a broader view of CP functionality may provide us with clues to optimize this therapy for COVID-19 and for future pandemics.

Two recent studies of anti-RBD monoclonal antibodies have shown that Fc-effector functions are essential for the therapeutic benefit in mice and hamster models of COVID-19[48,49]. Likewise, our data show that specific antibody effector functions contribute to antiviral immunity. Notably, S-specific ADNP activity was inversely correlated with the evolution of inflammatory SARS-CoV-2 antibodies in patients with low pre-existing antibody titers. Though typically thought of as primarily responsive to bacterial infections, neutrophils can also be the first responders to viral infections[50,51]. Our data suggest that enhancing the phagocytic capabilities of neutrophils, the first responding cells in COVID-19, may increase viral clearance before monocyte infiltration, which is implicated in COVID-19 hyperinflammation[52–54]. This non-neutralizing Fc-driven effect of CP may dampen inflammation and could in part explain the efficacy of early administration of CP to outpatients and those who are seronegative[19].

In randomized control trials, CP resulted in a 73% risk reduction of COVID-19 progression when given within 3 days of symptoms[19], but has had a mixed mortality benefit in randomized clinical trials when given later in illness to severe COVID-19 patients[21]. Similar results were obtained with monoclonal antibody therapies[39] and many have now focused on using antibody therapeutics in seronegative COVID-19. CP antibodies may have to compete with existing antibodies for Fc-receptor and lectin binding on immune cells. In patients with high pre-existing S-specific antibodies, this competition may be insurmountable for the amount of antibody in 1–2 units of CP and may explain why

signals of efficacy are mainly observed when CP is given to seronegative patients[19,55]. However, if functionally optimized, even small amounts of antibodies may be sufficient to tip the balance of pre-existing antibody pools. Quite interestingly, in individuals with high pre-existing S antibody levels, we found multiple N-specific antibody features in CP tracked with dampened inflammatory evolution of SARS-CoV-2-specific humoral immunity in CP recipients. The antibody responses to nucleocapsid, a highly expressed immunodominant antigen[56,57], can precede S-specific antibodies in some patients[58] and has been associated with progression to severe disease when involved in robust complement fixation[30]. Thus, the data presented here suggest less inflammatory CP-derived N-specific antibodies may play an indirect role in displacing patient-derived antibodies that contribute to enhanced pathology rather than immune protection. Further, the immunomodulatory effect of N-specific antibodies in CP may explain the mortality benefit seen in a small study of CP in severe COVID-19[23].

Altogether, the interplay of CP antibody profiles and severe COVID-19 patients antibody profiles demonstrates the critical role of pre-existing humoral immunity and the immunomodulatory activity of antibodies as therapeutics. Unexpectedly, CP was shown to modulate the evolution of inflammatory humoral immune responses via both S- and N-specific humoral immune responses, invoking a role for multiple lines of attack on the viral infection that could help support disease resolution. Further, these data suggest unique flavors of antibody therapeutics may be needed at different stages of COVID-19. Specifically, in the patients with low pre-existing S-IgG1 titers, an antibody therapeutic with strong neutralization and S-specific ADNP activity may have optimal antiviral and anti-inflammatory effects. On the other hand, patients with high pre-existing S-IgG1 titers may benefit from an antibody therapeutic with less inflammatory S antibodies and more immunomodulatory N-specific antibodies. Further antibody therapies enriched in N-specific antibodies in CP could be a new approach to alleviate the hyperinflammation of severe COVID-19 for which we have limited current therapies.

This study of patients treated with CP via the Mayo Clinic expanded access program was conducted early in the first wave of the COVID-19 pandemic in New York City. As there was no CP-untreated control, we cannot address the effect of CP on clinical outcomes or control for the natural evolution of the humoral immune response. Since the majority of enrolled patients already received corticosteroids prior to CP administration, we cannot address the effect of corticosteroid therapy on response to CP,

however, future large cohort-based studies should address the multifaceted effects of these drugs on both the adaptive and innate immune system. Further, our analysis identified different effects of CP on recipients with low or high pre-existing S-IgG antibodies. However, it remains unclear whether the CP recipients with low pre-existing S-IgG patients were earlier in the disease process or had a less fulminant disease. We relied on patient reported onset of symptoms as an approximation for their duration of illness. Despite these limitations, this study points to striking heterogeneity in CP and unexpected, distinct Fc-mediated humoral modulatory functions that may temper the evolution of the inflammatory response in severe COVID-19 disease. Our findings provide novel mechanistic insights into the impact of CP and provide unique hints for the rational design of next-generation monoclonal therapeutics with a longer-window of therapeutic efficacy and strategies to collect CP that may provide customized benefit based on CP recipient pre-existing antibody levels. In March 2020, CP was thought of as a therapy of necessity that would be replaced by more refined monoclonals. The rapid emergence of COVID-19 variants of concern is creating new therapeutic gaps that rationally designed and deployed CP may be needed to fill.

## Methods

**Study approval and informed consent**. The clinical cohort described in Yoon et al.[37] was approved by the Albert Einstein College of Medicine Institutional Review Board. Informed consent was obtained from all donors and the primary outcomes were published in Yoon et al.[37]. Secondary Use of patient samples and clinical samples was approved by the Mass General Brigham Institutional Review Board.

**Antibody titer and Fc-receptor binding assays**. Antigen-specific antibody subclass, isotype, and Fc-receptor (FcR) binding levels were assayed with a customized multiplexed Luminex bead array[59]. This allowed for relative quantification of antigen-specific humoral responses in a high-throughput manner and simultaneous detection of many antigens. A panel of SARS-CoV-2 antigens including the full spike glycoprotein (S) (provided by Lake Pharma), receptor-binding domain (RBD) (Provided by Aaron Schmidt, Ragon Institute) nucleocapsid (N) (Aalto Bio Reagents, Dublin, Ireland), S1 (Sino Biological, Beijing, China) and S2 (Sino Biological, Beijing, China) were used. Control antigens were run including a mix of three Flu-HA proteins (H1N1/A/New Caledonia/20/99, H1N1/A/Solomon Islands/3/2006, H3N2)(A/Brisbane/10/2007—Immune Tech) and Ebola glycoprotein (IBT Bioservices). Antigens were coupled to uniquely fluorescent magnetic carboxyl-modified microspheres (Luminex Corporation, Austin, TX) using 1-Ethyl-3-(3-dimethylaminopropyl) carbodiimide (EDC) (ThermoFisher Scientific, Waltham, MA) and Sulfo-N-hydroxysuccinimide (NHS) (ThermoFisher Scientific, Waltham, MA). Antigen-coupled microspheres were then blocked, washed, and incubated for 16 h at 4 °C while rocking at 700 rpm with diluted plasma samples at plate concentrations of 1:12,000 for all subclasses and isotypes and C1q and Fcrn binding and 1:120,000 for all other Fc-receptors to form immune complexes in a 20 μL volume in 384-well plates (Greiner, Monroe, NC). The following day, plates were washed using an automated plate washer (Tecan, Männedorf, Zürich, Switzerland) with 0.1% BSA and 0.02% Tween-20. Antigen-specific antibody titers were detected with Phycoerythrin (PE)-coupled antibodies against IgG1, IgG2, IgG3, IgG4, IgA1, and IgM (SouthernBiotech, Birmingham, AL). To measure antigen-specific Fc-receptor binding, biotinylated Fc-receptors (FcR2AH, 2B, 3AV, 3B, FCRN, FCAR, FCR3AV—Duke Protein Production facility, C1q—Sigma Aldrich) were coupled to Strepavidin-PE to form tetramers and then added to immune-complexed beads to incubate for 1 h at room temperature while shaking. Fluorescence was detected using an Intellicyt iQue with a 384-well plate handling robot (PAA) and analyzed using Forecyt software by gating on fluorescent bead regions. PE median fluorescence intensity (MFI) was measured as the readout of each antigen-specific antibody measurements. All experiments were performed in duplicate while operators were blinded to study group assignment and all cases and controls were run at the same time to avoid batch effects. The mean value of the duplicate measurements was used for further statistical analysis.

**Ab-directed functional assays**. Bead-based assays were used to quantify antibody-dependent cellular phagocytosis (ADCP), antibody-dependent neutrophil phagocytosis (ADNP), and antibody-dependent complement deposition (ADCD), as previously described[60–64]. Yellow (ADNP and ADCP), as well as red (ADCD) fluorescent neutravidin beads (ThermoFisher), were coupled to biotinylated SARS-CoV-2S antigens and incubated with diluted plasma (ADCP 1:100, ADNP 1:50, ADCD 1:10) to allow immune complex formation for 2 h at 37 °C. To assess

the ability of sample antibodies to induce monocyte phagocytosis, THP-1s (ATCC) were added to the immune complexes at 1.25E5 cells/mL and incubated for 16 h at 37 °C. For ADNP, primary neutrophils were isolated via negative selection (Stemcell) from whole blood. Isolated neutrophils at a concentration of 50,000 per well were incubated with immune complexes for 1 h incubation at 37 °C. Neutrophils were stained with an anti-CD66b PacBlue detection antibody (Biolegend) and fixed with 4% paraformaldehyde (Alfa Aesar). To measure the antibody-dependent deposition of C3, lyophilized guinea pig complement (Cedarlane) was reconstituted according to the manufacturer's instructions and diluted in gelatin veronal buffer with calcium and magnesium (GBV++) (Boston BioProducts) and mixed with immune complexes. After a 20-minute incubation at 37 °C, C3 was detected with an anti-C3 fluorescein-conjugated goat IgG fraction detection antibody (Mpbio). Antibody-dependent NK (ADNK) cell activity was measured via an ELISA-based assay, as described previously[35]. Briefly, plates were coated with 3 mg/mL of antigen (SARS-CoV-2S) and blocked overnight at 4 °C. NK cells were isolated the day of the assay with negative selection (RosetteSep - Stem Cell Technologies) from healthy buffy coats (MGH blood donor center). Diluted plasma samples were added to the antigen-coated plates (1:25 dilution) and incubated for 2 h at 37 °C. NK cells were mixed with a staining cocktail containing anti-CD107a BV605 antibody (BioLegend), Golgi stop (BD Biosciences), and Brefeldin A (BFA, Sigma Aldrich). 2.5E5 cells/mL were added per well to the immune complexes and incubated for 5 h at 37 °C. Next, cells were fixed (Perm A, Invitrogen) and stained for surface markers with anti-CD3 APC-Cy7 (BioLegend) and anti-CD56 PE-Cy7 (BD Biosciences). Subsequently, cells were permeabilized using Perm B (Invitrogen) and intracellularly stained with an anti-MIP-1β-BV421 (BD Biosciences) and IFNγ-PE (BioLegend) antibodies.

All assays were acquired via flow cytometry with iQue (Intellicyt) and an S-Lab 384-well plate handling robot (PAA). For ADCP, events were gated on singlets and bead-positive cells. For ADNP, neutrophils were defined as CD66b positive events followed by gating on bead-positive neutrophils. A phagocytosis score was calculated for ADCP and ADNP as (percentage of bead-positive cells) × (MFI of bead-positive cells) divided by 10000 as depicted in Supplementary Figs. 7 and 8. For ADCD, complement deposition was reported as the median fluorescence intensity of C3 deposition on Spike-coupled beads. For ADNK, NK cells were defined as CD3- and CD56 + events as depicted in Supplementary Fig. 6. NK cell activation was quantified as the percentage of NK cells positive for the degranulation marker CD107a[65] and for two markers of NK cell activation, MIP-1β, and IFNγ[66], as depicted in Supplementary Fig. 9. In the text, we referred to these readouts as CD107aNK, MIP-1βNK, and IFNγNK.

**rVSV-SARS-CoV-2S neutralization assay**. The neutralization assay was done as previously described[67]. Briefly, CP samples were serially diluted and incubated with pre-titrated amounts of virus for 1 h at room temperature, plasma-virus mixtures were added to 96-well plates (Corning) containing monolayers of Vero cells, incubated for 7 h at 37 °C/5% CO$_2$, fixed with 4% paraformaldehyde (Sigma) in PBS, washed with PBS, and stored in PBS containing Hoechst-33342 (1:2000 dilution; Invitrogen). Viral infectivity was measured by automated enumeration of green fluorescent protein (GFP)-positive cells from captured images using a Cytation5 automated fluorescence microscope (BioTek) and analyzed using the Gen5 data analysis software (BioTek). The serum half-maximal inhibitory concentration (IC50) was calculated using a nonlinear regression analysis with GraphPad Prism software. Neutralization titers were expressed as NT50 values (1/IC50).

**Statistics and reproducibility**. Duplicate measurements of antibody isotypes, subclasses, FcR-binding levels, and ADCD measurements were averaged for each sample and then log10 transformed. Duplicate measurements of ADNK, ADCP, and ADNP were averaged for each sample. To remove antibody features with low magnitude signals, we used the variation in the negative control samples as a cutoff, removing antibody features whose maximum signal in the CP recipients and CP donors was less than four standard deviations over the negative control PBS wells (Mean PBS + 4× PBS standard deviation). Using this pre-processing technique, we excluded three antibody features: nucleocapsid FCR2AH, S1 FCRN, and S1 C1q.

**Polar plots**. The data processing steps for both individual sample and group comparison polar plots are depicted in Supplementary Fig. 2A. Briefly, polar plots for Figs. 2, 3, and 4 were used to visualize the mean percentile of groups. Percentile rank scores were determined for each feature across all considered samples using the function "percent_rank" of the R package "dplyr" (1.0.5).

Polar plots for Fig. 1 and Supplementary Fig. 2B were used to visualize the S-specific individual antibody profile of CP units and pre-CP severe acute COVID-19 patients (day −1), respectively. Each feature across the respective populations was scaled by min–max normalization.

**Non-parametric combination**. Global statistical differences of feature types (e.g., IgG1) across antigens and between groups were assessed using non-parametric combination[68,69]. For each feature type, partial test p-values were obtained by Mann–Whitney U test for the comparison of day −1 and segment, and Friedman

tests for the comparison of day −1, day 1, and day 3 measurements for each antigen-specific sub-feature (e.g., S-IgG1, N-IgG1). The partial p-values were combined using Fisher's method to obtain a global statistic. This procedure to obtain a global statistic was repeated 1000 times for data with permuted group labels, preserving the permuted structure for the partial tests, and was used to construct a null distribution of global statistics. For the analysis of day −1, day 1, and day 3, only patients were included that had measurements at each time point, and for the permutations, the labels were shuffled for each patient individually. The true global statistic obtained from the unpermuted data was compared to the null distribution and a global p-value was determined as the tail probability. For the functions which were only measured for SARS-CoV-2S, the global p-value was obtained by merging all the functions.

**Testing for the effect of CP on humoral antibody trajectories with non-parametric combination testing of the median difference of spearman correlation**. To evaluate the significance of correlation difference between high S-IgG1 and low S-IgG1 group, we permuted the group of the recipient–donor pairs with the fixed proportion of high and low S-IgG1 group 800 times and then calculated the median correlation of each feature with donor CP features. After that, we estimated the p-values as the proportion of permutated median correlation values of permutations above and below the observed the actual median. The features with a permutation test p-value of correlation difference less than 0.1 along with S-IgG1 and Neutralizing antibody titer were selected for further exploration.

To evaluate the effect of recipient–donor pair, global statistical differences of each feature in CP across all the features in recipient were evaluated using non-parametric combination as described above. For the high S-IgG1 or low S-IgG1 group, respectively, we broke the recipient–donor pair, randomly matched them, and calculated the Spearman correlation on the permuted recipient–donor pair 1000 times. The partial test p-values were obtained by Spearman correlation and were combined using Fisher's method to obtain a global statistic by comparing actual value from the true recipient–donor pairs with the null distribution generated from permuted pairs for each feature in CP.

**Multivariate models**. Partial least square discriminant analysis (PLS-DA) was performed to discriminate day −1 patient samples from donor CP samples. Multi-level PLS-DA (mPLS-DA)[70] was performed to discriminate day −1 and day 1/day 3 measurements and to take into account the paired structure of the data. For the multi-level PLS-DA, only patients who had measurements at both considered time points were included. Missing values were imputed using k-nearest neighbor imputation (R package "DMwR" (0.4.1)) before z-scoring. For the mPLS-DA, the data was first imputed, denoised, and then z-scored. The model performance was assessed in a five-fold cross-validation framework, and the average cross-validation accuracy was reported for 100 repetitions of cross-validation. For the mPLS-DA, the folds for cross-validation are generated such that measurements of the same patient are in the same fold. Variable importance of projection (VIP) scores, which describe the contribution of each feature to the model, were used to rank the features, and the top 10 important features were displayed. The modeling approach was validated using permutation tests. Control models with "permuted labels" were generated, for which the model was trained and applied to data with shuffled group labels in the same cross-validation framework. For the case of paired data (each patient has measurements at both time points) in the mPLS-DAs, the labels were flipped with a 50% probability to obtain the control models. This procedure was done for 500 permutations for each of 10 cross-validation replicates. The p-values for the modeling approach were obtained from the tail probability of the generated null distribution, i.e., the distribution of classification accuracies of the control models. The median p-value across the 10 cross-validation replicates was reported in the figures. PLS-DA models were generated with the R package "ropls" (1.22.0) interfaced by R package "systemsseRology" (https://github.com/LoosC/systemsseRology). The analyses were performed with R version 4.0.2.

**Correlation analysis**. We calculated Spearman correlation and their p-values using the R function "cor.test" of the R package "stats" (4.0.3). After that, the p-values were adjusted by Benjamini–Hochberg procedure for multiple testing correction. The adjusted p-values were labeled by asteriks (*: adjusted p-value < 0.05) in the correlation heatmap if they were significant.

A chord diagram was used to visualize the significant links among the humoral features using the function "chordDiagram" in R package "circlize" (0.4.12).

To evaluate co-correlated relationships between top features selected by PLS-DA and additional humoral immune features, the significant Spearman correlations above a threshold of |r| > 0.75 and p-value < 0.05 were selected and a layout was created to specify the spatial position to maintain correlation patterns using the function "create_layout" in R package "ggraph" (2.0.4), where the gradient color of links represented the strength of the correlations and the color of nodes denoted the enriched group. After that, the layout was visualized as the correlation network using the function "ggraph" in the R package "ggraph" (2.0.4). Additionally, the labels and positions of nodes and links were adjusted for better visualization using the software Adobe Illustrator 2020 (24.2.3).

**Bootstrapping estimation of variation in percentile rank**. To estimate the variance of the mean of percentile rank per each measurement across the patients at day −1 and convalescent plasma, a stratified bootstrapped sampling strategy with replacement was applied here 1000 times. Then, we calculated the mean of percentile for each antibody feature in both clinical groups. The mean percentile and 95% confidence interval were then visualized as a boxplot using the R package ggplot2 (3.3.5).

**Reporting summary**. Further information on research design is available in the Nature Research Reporting Summary linked to this article.

## Data availability

The data set generated during and/or analyzed during the current study has been made available in Supplementary Data 1. A Source Data file containing the raw numbers for the figures are also provided. No data was stored externally. Source data are provided with this paper.

## Code availability

Custom code was used in this manuscript and has been made available at https://zenodo.org/record/5527197#.YU42RrhKiUl. The R packages used for data analysis are described in more detail in the "Methods" section and more information is available upon request.

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

## Acknowledgements

We thank Nancy Zimmerman, Mark, and Lisa Schwartz, an anonymous donor (financial support), Terry and Susan Ragon, and the SAMANA Kay MGH Research Scholars award for their support. We acknowledge support from the Ragon Institute of MGH, MIT, and Harvard, the Massachusetts Consortium on Pathogen Readiness (MassCPR), the NIH (3R37AI080289-11S1, R01AI146785, R01-AI132633, R01-AI132633, U19AI42790-01, U19AI135995-02, U19AI42790-01, 1U01CA260476 – 01, CIVIC75N93019C00052, T32 AI007061, R01AI123654, R01AI143453, 3UL1TR002556-04S1), the G. Harold and Leila Y. Mathers Foundation, the Gates Foundation, the Global Health Vaccine Accelerator Platform funding (OPP1146996 and INV-001650), the Musk Foundation, and the Price Family Foundation.

## Author contributions

J.D.H., B.J., K.J.B., D.L., L.A.P, and G.A. conceived of the idea. H.Y., J.R., M.E.D., D.H., R.K.J., R.H.B. II, K.C., and L.A.P. designed, conducted, analyzed the clinical cohort, and performed neutralization antibody titer measurements. J.D.H. and G.A. designed the experiments. J.D.H. performed the experiments except for the neutralization antibody titer measurements. J.D.H., C.W., D.L, and C.L. analyzed the data. And J.D.H., C.W., C.L., D.L., L.-a.P, and G.A. wrote the paper with input from all authors.

## Competing interests

G.A. is a founder of SeromYx Systems, Inc. and an equity holder in Leyden Labs. G.A. is a member of the scientific advisory board of Sanofi Pasteur. K.C. is a member of the scientific advisory boards of Integrum Scientific LLC, Biovaxys Technology Corp, and the Pandemic Security Initiative of Celdara Medical, LLC. K.C., M.E.D., D.H., and R.K.J., are named co-inventors on a patent application assigned to Albert Einstein College of Medicine that covers a SARS-CoV-2 surrogate neutralization assay. The remaining authors declare no competing interests.
