## [Peer Review File · Nature Communications]

REVIEWER COMMENTS

Reviewer #1 (COVID-19, convalescent plasma therapy) (Remarks to the Author):

The article "Functional Antibodies in COVID-19 Convalescent Plasma" compares the quantitative and qualitative differences of SARS-CoV-2 specific antibodies in acute COVID-19 patients and the COVID-19 convalescent plasma (CCP).

The aim, results and methods of the article are of interest. Antigen-specific antibody titers and antigen-specific Fc receptor binding were measured. The authors did functional studies, such as antibody-dependent cellular phagocytosis (ADCP), neutrophil phagocytosis (ADNP), and complement deposition (ADCD). These methods are of interest and differ from other studies.

SARS-CoV-2-specific Fc heterogeneity across CCP units, and their recipients, was stressed in the article. The conclusion highlights the anti-inflammatory and direct antiviral effects of convalescent plasma.

Below are the comments on the manuscript:

1. The authors wrote convalescent plasma (CP) as CCP in the first sentence of the text. It may be better CP.
2. The abstract may better mention the methods, such as ADCP and ADCD, at least namely.
3. Page 4, Line 107: Please define or cite 'System Serology'.
4. The last paragraph (Line 109-116) of the 'Introduction' includes the results. The authors should give the results in the 'Results' section.
5. The authors should give the Results section in a more configured way by mentioning the The authors Figures in Results
6. Page 5, Line 121: Trattreat
7. Page 5, Line 122: 3 days  three days
8. Page 5, Line 134: WHO ordinal scale value?? Explain.
9. Page 6, Line 157-8: The result should not include a comment.
10. Page 7, Line 188: Please use the abbreviation 'RBD-specific IgM' after explained.
11. Typo:
Page 11, Line 285: has not 'be' linked  has not been linked.
12. Typo:
Page 14, Line 356: Sever severe
13. The authors should shorten the first paragraph of the 'Discussion' to discuss the novel findings. They stressed the heterogeneity of the CP. However, other articles are mentioning this issue (e.g. Expert Rev Clin Immunol. 2021 Mar 12 :1-7. doi: 10.1080/1744666X.2021.1894927.) that the authors may refer to).
14. Page 11, Line 286: Why did the authors use 'non-pathogen specific antibodies in this sentence?

Reviewer #2 (Viral immunity, bnAb) (Remarks to the Author):

Summary:

Herman et al. describe the deep function profiling of the antibody response in 19 severe COVID-19 patients before and after administration of convalescent plasma (CCP) and compare these functional profiles with those in the 14 different CCP units administered to the patients. They observed striking SARS-CoV-2 specific Fc-heterogeneity across CCP units and their recipients. However, CCP units possessed more functional antibodies than acute COVID-19 and link the influence of both S and Nucleocapsid (N) specific antibody functions not only in direct antiviral activity but also in anti-

inflammatory effects. Though CCP is currently not the therapeutic of choice in most clinical situations, it remains valuable to further assess how donor antibodies can influence the development of functional antibodies in CPP recipients. Overall, the manuscript is well-written and the experiments and analysis described in the manuscript are of high standard. The major concern is the lack of control patients, see specific comments below.

Major comments:

This study is missing context from relevant controls; severe COVID-19 patients who did not receive CPP. It would be very informative to be able to assess the natural evolution of antibody functionality to the response observed in the CPP recipients. Was the effect observed in this study, truly because of the CPP treatment or could the natural evolution of the antibody response in patients result in similar observations. Additionally, it would be informative if the authors described if differences are detectable between patients with different timing of CPP administration (based on days since hospitalization and/or time since symptom onset).

Minor comments:

1. The abstract would be more informative if more words are spent on describing the results and less describing the background. The relevance of the results could be formulated more clearly. Additionally, the sentence starting on line 62 is not well formulated.
2. A clearer explanation for why systems serology on CCP recipients is useful and why these different features are chosen to be analyzed might add to the introduction.
3. Since for one patient it could not be identified which CCP unit was administered, exclusion of this patient from the study might be more appropriate.
4. Describing proportions for a total of 19 patients is more clear if the actual numbers are reported instead of percentages.
5. The altered order of the polar plots of the CCP recipients in figure 1a is confusing. Since it is unclear what the added value of the hierarchical clustering is, it would be clearer to restore the numerical order.
6. The p value which was considered statistically significant is missing in the legend of figure 1b.
7. Figure 1a and 1b should have the same orientation of the different features.
8. The figures can be interpreted better when the different features and why they are analyzed are explained somewhere in the text or figure legend. Examples: (1) The text uses the abbreviation ADNK (line 181), but this feature is called differently in the figures. (2) it is not explained what the abbreviation S2P, used in figure legend 2 stands for.
9. In the manuscript, there are several unscientific, contradicting or exaggerative terms which should be replaced with more suitable expressions. Examples: striking yet expected (line 146); expansion (line 166); significant (line 177); explosion (line 206).; unique (line 249).
10. In figure 2a and 2b the plots for CPP units are called segment which is very confusing.
11. Due to the large heterogeneity displayed in figure 1 and supplementary figure 2, the interpretation of figure 2a would be greatly enhanced by the use of error bars or another way to describe the variation in these averages.

12. The added value of figure 2D is not very clear, though it is elaborately described in the results. The main conclusion of the results describing figure 2 does not need the analysis of figure 2D so perhaps it may be removed.

13. Supplementary figure 4 and 5 A-E are not referred to in the manuscript text.

14. Except for figure 3a and supplementary figure 3, the figures do not show the actual data, and therefore it is very hard to interpret what the actual signal strength of the antibody binding and functions was. It might be clearer how to interpret the data when the first figure or supplement for the first figure explains the type of data used.

15. In the results section, the conclusions would be more appropriate if kept a bit more modest. Also it could be more clearly indicated when speculating for possible meaning or effect of the findings.

16. The discussion is overly long and elaborate. Paragraph 2 would be more fitting in an introduction. Paragraph 3 is repetitive with the results and would be better if discussions points were incorporated. Paragraph 4 would be better suited in 2-3 sentences. Paragraph 5 and 6 are interesting but may be formulated a bit more concise.

REVIEWER COMMENTS

Reviewer #1 (COVID-19, convalescent plasma therapy) (Remarks to the Author):

The article "Functional Antibodies in COVID-19 Convalescent Plasma" compares the quantitative and qualitative differences of SARS-CoV-2 specific antibodies in acute COVID-19 patients and the COVID-19 convalescent plasma (CCP).

The aim, results and methods of the article are of interest. Antigen-specific antibody titers and antigen-specific Fc receptor binding were measured. The authors did functional studies, such as antibody-dependent cellular phagocytosis (ADCP), neutrophil phagocytosis (ADNP), and complement deposition (ADCD). These methods are of interest and differ from other studies.

SARS-CoV-2-specific Fc heterogeneity across CCP units, and their recipients, was stressed in the article. The conclusion highlights the anti-inflammatory and direct antiviral effects of convalescent plasma.

Below are the comments on the manuscript:

1. The authors wrote convalescent plasma (CP) as CCP in the first sentence of the text. It may be better CP.

Response: We thank the reviewers for this insight. We have changed the abbreviation throughout the manuscript to make it more consistent with the rest of the field

2. The abstract may better mention the methods, such as ADCP and ADCD, at least namely.

Response: We appreciate the reviewer's suggestions to improve our abstract to better highlight the unique methods used in this work. We have made changes to incorporate the methodology in the abstract, results, and methods sections.

3. Page 4, Line 107: Please define or cite 'System Serology'.

Response: We thank the reviewer for identifying this oversight. We have added a citation and more explanation of Systems Serology / functional antibody profiling methodology in the Introduction of the manuscript.

4. The last paragraph (Line 109-116) of the 'Introduction' includes the results. The authors should give the results in the 'Results' section.

Response: We appreciate the reviewer's comments and have edited the introduction to limit the results data disclosed.

5. The authors should give the Results section in a more configured way by mentioning the The authors Figures in Results

Response: We have added direct mention of the figures throughout the Results section.

6. Page 5, Line 121: Trattreat

7. Page 5, Line 122: 3 days  three days

Response: We apologize for the typos and have corrected them.

8. Page 5, Line 134: WHO ordinal scale value?? Explain.

Response: We thank the reviewer's for identifying our oversight. We have added a citation and explanation of the WHO ordinal scale value of 5.

9. Page 6, Line 157-8: The result should not include a comment.

Response: We appreciate the reviewer's comments and we have removed the commentary from this summary sentence.

10. Page 7, Line 188: Please use the abbreviation 'RBD-specific IgM' after explained.

Response: We appreciate the reviewer's comments and we have altered the abbreviations as described.

11. Typo:

Page 11, Line 285: has not 'be' linked  has not been linked.

12. Typo:

Page 14, Line 356: Sever severe

Response: We thank the reviewer for pointing out these typos and we have corrected them.

13. The authors should shorten the first paragraph of the 'Discussion' to discuss the novel findings. They stressed the heterogeneity of the CP. However, other articles are mentioning this issue (e.g. Expert Rev Clin Immunol. 2021 Mar 12 :1-7. doi: 10.1080/1744666X.2021.1894927.) that the authors may refer to).

Response: We revised the discussion extensively and added the reference the reviewer cited.

14. Page 11, Line 286: Why did the authors use 'non-pathogen specific antibodies in this sentence?

Response: We thank the reviewer for this question. We have revised the phrase to point out that IVIG used in the study we cited is a pool of antibodies collected from multiple pre-pandemic donors to emphasize that the effect of IVIG is not because of anti-SARS CoV2 antibodies but rather non-CoV2 specific antibody effects.

Reviewer #2 (Viral immunity, bnAb) (Remarks to the Author):

Summary:

Herman et al. describe the deep function profiling of the antibody response in 19 severe COVID-19 patients before and after administration of convalescent plasma (CCP) and compare these functional profiles with those in the 14 different CCP units administered to the patients. They observed striking SARS-CoV-2 specific Fc-heterogeneity across CCP units and their recipients. However, CCP units possessed more functional antibodies than acute COVID-19 and link the influence of both S and Nucleocapsid (N) specific antibody functions not only in direct antiviral activity but also in anti-

inflammatory effects. Though CCP is currently not the therapeutic of choice in most clinical situations, it remains valuable to further assess how donor antibodies can influence the development of functional antibodies in CPP recipients. Overall, the manuscript is well-written and the experiments and analysis described in the manuscript are of high standard. The major concern is the lack of control patients, see specific comments below.

Major comments:

This study is missing context from relevant controls; severe COVID-19 patients who did not receive CPP. It would be very informative to be able to assess the natural evolution of antibody functionality to the response observed in the CPP recipients. Was the effect observed in this study, truly because of the CPP treatment or could the natural evolution of the antibody response in patients result in similar observations. Additionally, it would be informative if the authors described if differences are detectable between patients with different timing of CPP administration (based on days since hospitalization and/or time since symptom onset).

Response: We agree with the reviewer, a control group of patients who had not received CP is necessary to discriminate the effect of CP on the evolution of CP recipient antibody profiles from the natural response. However, there was no control group. The cohort we studied was treated with CP using the Mayo Clinic Expanded Access Program protocol¹. It was an open label study conducted at the height of the first pandemic wave in Bronx, NY². The patients were the first patients in the Bronx, and many of the first in the US to receive CP. Knowledge gained from their treatment is important and may inform practices if there is another COVID-19 wave or another pandemic for which there is no therapy. In this regard, our data provide valuable new insights into the nature of CP and evolution of the CP recipient antibody responses of people who receive it. Though we cannot directly ascribe the differences we observed to CP without a control group, differences in antibody profiles in patients with high and low titer SARS-CoV-2 antibody levels before CP was administered suggest it may have shaped the response differently, in addition to having immunomodulatory effects.

We now discuss these limitations in the discussion and have planned a new study to compare profiles in CP recipients who received CP or placebo in a randomized controlled trial to address the very question the reviewer addresses.

We thank the reviewer for their point about the time of onset of disease and the effect of the timing of natural infection on anti-COVID-19 antibody profiles. To address this question, we evaluated the relationship of high and low pre-existing (day -1) S-IgG1 to the time of disease onset. The result is here:

Supplementary Figure 4A:

S-IgG1 titer for each of the 19 patients plotted by duration of COVID-19 symptoms. Patients with high pre-existing (day -1) S-IgG1 titers are shown in pink and those with lower pre-existing S-IgG1 titers are shown in yellow.

This new figure (**Supplementary Figure 4A**) shows that high day -1 titers are not surrogates for longer disease duration. In fact, many with shorter symptom durations had higher titers, suggesting they may have had fulminant or more rapidly progressive disease.

To ask how timing of CP administration may have changed the effect of CP on evolution of the antibody response, we grouped the patients by symptom duration at the time of CP administration and performed Spearman correlations to seek relationships between CP Ab features and patient antibody trajectories as a function of symptom duration. We then created a subtraction matrix of the two correlation matrixes, identified the median difference of correlation for each CP-provided antibody feature, and used permutation testing to assess for significance. This analysis plan mirrored our approach to the high and low pre-existing titer analysis in Figure 4.

This analysis showed that, unlike the pre-existing Ab titer analysis, the correlations with the CP-nucleocapsid features were quite varied, the differences in correlations were very heterogenous, and in some cases data were discrepant (e.g. Spike FCaR is divergent in the two groups but Spike IgA1 is not). We plan to revisit this approach in a future study with a larger dataset and controls. Therefore, in this manuscript we plan to keep the focus of our analysis of CP-induced changes in patient humoral profiles dependent on pre-existing S-IgG titer analysis in our current Figure 4.

Minor comments:

1. The abstract would be more informative if more words are spent on describing the results and less describing the background. The relevance of the results could be formulated more clearly. Additionally, the sentence starting on line 62 is not well formulated.

Response: We appreciate the reviewer's point and modified the abstract as suggested. We also rewrote the sentence in (former) line 62.

2. A clearer explanation for why systems serology on CCP recipients is useful and why these different features are chosen to be analyzed might add to the introduction.

Response: We appreciate this comment and now provide a more extensive explanation of Systems Serology and why it is useful in studying CP, the main point being that it can reveal the importance of antibodies that work via Fc receptors and have the potential to modulate inflammation.

3. Since for one patient it could not be identified which CCP unit was administered, exclusion of this patient from the study might be more appropriate.

Response: We thank the reviewer for this suggestion. The data we show in this manuscript is part of an exploratory study of hospitalized patients conducted at the height of the COVID-19 pandemic in New York City. Unfortunately under these conditions, we were not able to obtain the CP unit that one patient received. Thus, we have excluded him/her from all analysis of the interaction of CP unit with CP recipients.

However, we did include this patient in the part of our analysis that did not require the mapping of CP unit to CP recipient. This included the analysis that only compared disease states (convalescents heterogeneity in Figure 1, convalescents vs. severe COVID-19 in Figure 2) and the analysis of the antibody trajectories in CP-treated severe COVID-19 (Figure 3). We believe this is a reasonable compromise to try to present the most robust description of humoral antibody profiles in this hypothesis generating human subjects research.

4. Describing proportions for a total of 19 patients is more clear if the actual numbers are reported instead of percentages.

Response: We thank the reviewer for this suggestion. We changed the description of the cohort's clinical characteristics to the actual number of patients rather than percentages and made clear that outcomes were measured on day 28.

5. The altered order of the polar plots of the CCP recipients in figure 1a is confusing. Since it is unclear what the added value of the hierarchical clustering is, it would be clearer to restore the numerical order.

Response: We removed the hierarchical clustering of CP unit radial plots from Figure 1A.

7. Figure 1a and 1b should have the same orientation of the different features.

Response: We changed Figure 1A and 1B to have the same organization.

6. The p value which was considered statistically significant is missing in the legend of figure 1b.

Response: We added that a P-value of <0.05 was used.

8. The figures can be interpreted better when the different features and why they are analyzed are explained somewhere in the text or figure legend. Examples: (1) The text uses the abbreviation ADNK (line 181), but this feature is called differently in the figures. (2) it is not explained what the abbreviation S2P, used in figure legend 2 stands for.

Response: We thank the reviewer for this suggestion. We removed the S2P data from the figures because of its redundancy with Spike-specific antibody measurements. In addition, we included additional text to explain the ADNK features, NK cell expression of CD107a, IFN γ , and MIP1b, in the figure legends, the Results section, and the Methods section.

9. In the manuscript, there are several unscientific, contradicting or exaggerative terms which should be replaced with more suitable expressions. Examples: striking yet expected (line 146); expansion (line 166); significant (line 177); explosion (line 206).; unique (line 249).

Response: We thank the reviewer for this comment and revised all such comments. For example, for the original line 177, we have altered the text to clarify that the PLSDA model was validated with a 500-fold cross validation model and found to be statistically significant by permutation testing.

10. In figure 2a and 2b the plots for CPP units are called segment which is very confusing.

Response: We agree and we have changed the designation to "CP units."

11. Due to the large heterogeneity displayed in figure 1 and supplementary figure 2, the interpretation of figure 2a would be greatly enhanced by the use of error bars or another way to describe the variation in these averages.

Response: We appreciate the reviewer's comment and agree on the importance of displaying the heterogeneity of the data displayed in the radial plots in **Figure 2**. **Supplementary Figure 2E** displays the MFIs of each individual CP unit and day-1 sample for a subset of differential enriched antibody features. To further address this point, we have created a new **Supplementary Figure 2C** that shows the data in **Figure 2A** with error bars.

Supplementary Figure 2C

To estimate the variance in the percentage ranks of antibody features in the day -1 samples and CP units in Figure 2A, we used a stratified bootstrapping sampling strategy with replacement with 1000 replicates and graphed the mean and IQR of each antibody feature for each clinical group. Though there is heterogeneity, as indicated by error bars, this is surpassed by the magnitude of difference of Spike IgA1 titer and FCaR binding between recipient day -1 and CP unit samples.

12. The added value of figure 2D is not very clear, though it is elaborately described in the results. The main conclusion of the results describing figure 2 does not need the analysis of figure 2D so perhaps it may be removed.

Response: We appreciate the reviewer's point. We included **Figure 2D** to help show the relationships inherent in a polyclonal antibody response. While our PLS-DA model identified the most salient features that separate the group, we believe an appreciation of the polyclonal network is critical to identify features shared and unshared between CP units and CP recipients. We have retained the figure, clarified the point, and tightened the description.

13. Supplementary figure 4 and 5 A-E are not referred to in the manuscript text.

Response: We thank the reviewer for the comment. We added a description of **Supplementary Figure 4A** to the results. Since **Supplementary Figure 4B** and **Supplementary Figure 5** are not essential to our description of the results, we removed them from the revised manuscript.

14. Except for figure 3a and supplementary figure 3, the figures do not show the actual data, and

therefore it is very hard to interpret what the actual signal strength of the antibody binding and functions was. It might be clearer how to interpret the data when the first figure or supplement for the first figure explains the type of data used.

Response: We thank the reviewer for the comment. We added a new flow diagram as **Supplementary Figure 1A** that describes the data analysis steps from primary data to the two types of radial plots that make up **Figure 1** (Individual polar plot) and **Figures 2, 3, and 4** (Group Comparison Polar Plots).

Supplementary Figure 2A: Flow diagram of the data processing steps to make Individual Sample and Group Comparison polar plots.

We also included a new **Supplementary Figure 2C** that demonstrates the variation inherent in our antibody profiling data.

15. In the results section, the conclusions would be more appropriate if kept a bit more modest. Also it could be more clearly indicated when speculating for possible meaning or effect of the findings.

Response: We appreciate the reviewer’s thoughtful comments and have altered the results section in the following ways:

- We removed speculative language in the results section
- We indicated where we used speculation to interpret our results.

16. The discussion is overly long and elaborate. Paragraph 2 would be more fitting in an introduction. Paragraph 3 is repetitive with the results and would be better if discussions points were incorporated. Paragraph 4 would be better suited in 2-3 sentences. Paragraph 5 and 6 are interesting but may be formulated a bit more concise.

Response: We thank the reviewer for these comments and have revised the discussion extensively based on these suggestions.

1 Senefeld, J. W. *et al.* Program and patient characteristics for the United States Expanded Access Program to COVID-19 convalescent plasma. *Medrxiv*, 2021.2004.2008.21255115, doi:10.1101/2021.04.08.21255115 PMID - 33851175 (2021).

- 2 Yoon, H. a. *et al.* Treatment of Severe COVID-19 with Convalescent Plasma in Bronx, NYC. *Jci Insight*, doi:10.1172/jci.insight.142270 PMID - 33476300 (2021).

REVIEWERS' COMMENTS

Reviewer #1 (Remarks to the Author):

Dear Editor, Dear Authors,

The authors have studied the humoral features of convalescent plasma. In addition to the CP antibody profiling, antibody-related parameters were tested from 19 severely ill COVID-19 patients' plasma (pre-CP (-1st day) and post-CP (+1st and 3rd days)) to show the effect of the CP. The authors measured the neutralization titers, anti-N- antibody and anti-spike antibody and subtypes and isotypes of other antibodies, ADCC, ADNC, ADNK, and ADCD, as well as Fc receptor binding. Although correlation with some of the clinical parameters, such as the measurement of acute phase reactants, inflammatory cytokines of the patients during the CP therapy would strengthen the study, the design of the study is well when we take into account the huge variability of the convalescent plasma and treatment characteristics of patients.

The authors have considerably revised the manuscript according to the reviewers' suggestions. However, the manuscript still needs revision.

- 18 out of 19 patients are under corticosteroids therapy. The length (the initiation, cessation, etc.) and the dose of the corticosteroids therapy should be given in the manuscript. The effect of corticosteroids on ADCC, ADNP, ADCC, and antibody profiling may be discussed.

- Line 353: All experiments were performed in duplicate.....

Did the authors take the mean value of duplicate data during the statistical studies?

- Line 394:1hr at RT. RT?? Please explain the abbreviation.

- Line 406: Please control this sentence below:

Antibody features were excluded from the analysis if the maximal value across samples and time points was less than four standard deviations above the mean value obtained for PBS controls.

Typo:

1. Page 8, line 199: Explanation of the abbreviation 'RBD' is missing.

2. Page 11, line 270:

...both Spike and Nucleocapsid-specific both S- and N-specific

3. Page 11, line 267: ...infected cells via infected cells via complement....

4. Page 12, line 285:has not had a mixed mortality benefit...

Reviewer #2 (Remarks to the Author):

The authors have responded to all review comments and have made major revisions in the manuscript. There is still one point that could be interpreted differently that could influence their analysis and interpretation of the data significantly. When the authors looked at the relationship of high and low pre-existing (day -1) S-IgG1 to the time of disease onset, they interpret the high group as having fulminant or more rapidly progressive disease. However, the self-reporting of symptoms and start of symptoms is very diverse, as some people only report severe symptoms and others the smallest as they are differently experienced. Could it just be that the high group has the reported start of symptoms at a later stage of infection compared to the low group which results in already higher antibodies because of longer infection?

Response to Reviewers:

Reviewer 1:

The authors have considerably revised the manuscript according to the reviewers' suggestions. However, the manuscript still needs revision.

- 18 out of 19 patients are under corticosteroids therapy. The length (the initiation, cessation, etc.) and the dose of the corticosteroids therapy should be given in the manuscript. The effect of corticosteroids on ADCC, ADNP, ADCC, and antibody profiling may be discussed.

We agree with the reviewer about the importance of describing corticosteroid use in this cohort. We have included the median duration of steroids and number of patients who received CP before steroids in the main text and added the following to **Supplementary Table 1**: which steroids were used in the study, median steroid dose over admission, mean days of steroid therapy, and the mean daily steroid dose. We have added a comment in the discussion to address the effects of steroids on this set of experiments and the limited power we have to make any relevant observations.

- Line 353: All experiments were performed in duplicate.....

Did the authors take the mean value of duplicate data during the statistical studies?

We appreciate the reviewers point, and have added to the methods to describe that the mean of the duplicate measurements were used for further analysis.

- Line 394:1hr at RT. RT?? Please explain the abbreviation.

We have removed the abbreviation.

- Line 406: Please control this sentence below:

Antibody features were excluded from the analysis if the maximal value across samples and time points was less than four standard deviations above the mean value obtained for PBS controls.

We have re-written the sentence to better explain our data pre-processing methods.

Typo:

1. Page 8, line 199: Explanation of the abbreviation 'RBD' is missing.

We have added an abbreviation for RBD.

2. Page 11, line 270:

...both Spike and Nucleocapsid-specific both S- and N-specific

We have used the abbreviated form of Spike and Nucleocapsid

3. Page 11, line 267: ...infected cells via infected cells via complement....

We have removed the repeated phrase.

4. Page 12, line 285:has not had a mixed mortality benefit...

We have removed the inappropriate 'not' from the sentence.

Reviewer 2:

The authors have responded to all review comments and have made major revisions in the manuscript. There is still one point that could be interpreted differently that could influence their analysis and interpretation of the data significantly. When the authors looked at the relationship of high and low pre-existing (day -1) S-IgG1 to the time of disease onset, they interpret the high group as having fulminant or more rapidly progressive disease. However, the self-reporting of symptoms and start of symptoms is very diverse, as some people only report severe symptoms and others the smallest as they are differently experienced. Could it just be that the high group has the reported start of symptoms at a later stage of infection compared to the low group which results in already higher antibodies because of longer infection?

We agree with the reviewers. Based on our analysis, the level of pre-existing S-IgG1 antibody can alter the effect of CP. However, it remains unclear whether patients with low pre-existing S-IgG1 could be earlier in disease or have less fulminant disease. Unfortunately, we cannot discern between the two possibilities with our cohort. I have added this point to our discussion to clarify the reviewer's astute point.